# Obesity and Bone Health: A Complex Relationship

**DOI:** 10.3390/ijms23158303

**Published:** 2022-07-27

**Authors:** Ana Piñar-Gutierrez, Cristina García-Fontana, Beatriz García-Fontana, Manuel Muñoz-Torres

**Affiliations:** 1Endocrinology and Nutrition Division, University Hospital Virgen del Rocío, 41013 Sevilla, Spain; anapinarg@gmail.com; 2Bone Metabolic Unit, Endocrinology and Nutrition Division, University Hospital Clínico San Cecilio, 18016 Granada, Spain; 3Instituto de Investigación Biosanitaria de Granada (Ibs. GRANADA), 18012 Granada, Spain; 4Centro de Investigación Biomédica en Red Fragilidad y Envejecimiento Saludable (CIBERFES), Instituto de Salud Carlos III, 28029 Madrid, Spain; 5Department of Medicine, University of Granada, 18016 Granada, Spain

**Keywords:** obesity, fracture, body composition, inflammation, healthy aging, osteoporosis

## Abstract

Recent scientific evidence has shown an increased risk of fractures in patients with obesity, especially in those with a higher visceral adipose tissue content. This contradicts the old paradigm that obese patients were more protected than those with normal weight. Specifically, in older subjects in whom there is a redistribution of fat from subcutaneous adipose tissue to visceral adipose tissue and an infiltration of other tissues such as muscle with the consequent sarcopenia, obesity can accentuate the changes characteristic of this age group that predisposes to a greater risk of falls and fractures. Other factors that determine a greater risk in older subjects with obesity are chronic proinflammatory status, altered adipokine secretion, vitamin D deficiency, insulin resistance and reduced mobility. On the other hand, diagnostic tests may be influenced by obesity and its comorbidities as well as by body composition, and risk scales may underestimate the risk of fractures in these patients. Weight loss with physical activity programs and cessation of high-fat diets may reduce the risk. Finally, more research is needed on the efficacy of anti-osteoporotic treatments in obese patients.

## 1. Introduction

Obesity and osteoporosis are two very prevalent diseases in the older subjects. Both lead to increased morbidity and mortality and therefore have a high negative impact on public health worldwide [1]. On the one hand, osteoporosis is characterized by low bone mineral density (BMD) and an alteration of the microarchitecture of bone tissue, which leads to an increased risk of fractures [2]. According to WHO criteria, its diagnosis is based on BMD measurement with T-Score ≤ 2.5 SD values [3]. In the United States, it is currently the cause of about 500,000 hospitalizations, more than 2.6 million doctor visits, 800,000 emergency room admissions, and 180,000 nursing home admissions [4]. Given the aging population, its prevalence is expected to increase and by 2040 the associated cost is expected to rise by 100–200% [4]. Regarding obesity, it is a complex disease in which there is an increase in body weight and especially an excess of adipose tissue. The diagnostic criterion according to the WHO is a body mass index (BMI) ≥ 30 kg/m^2^. Its current prevalence is more than double that of 30 years ago, with an estimated 13% of adults worldwide being obese in 2016 [5]. This is due to lifestyle changes that have occurred in Western countries especially but also in Eastern countries in recent years, consisting of an increase in caloric intake with high-fat foods and a decrease in physical activity with a sedentary lifestyle. In addition to increasing health care spending by billions of dollars [6], it is a clear risk factor for diseases such as type 2 diabetes, high blood pressure, chronic kidney disease, cancer, coronary heart disease and cerebrovascular disease [5].

In particular, the relationship between obesity and type 2 diabetes is especially relevant, the first being the main risk factor of the latter. Type 2 diabetes is also a highly prevalent disease with a high socio-economic cost. It is estimated that in 2035 the global prevalence of type 2 diabetes will be 592 million people. As for its relationship with osteoporosis and fractures, patients with diabetes, although they usually have a normal or even high BMD, have a paradoxical increase in the risk of fracture [7]. Specifically, the risk of hip fracture is increased by 1.3 to 2.3 times, and of other fractures by 1.2 times, except in the case of vertebral fractures, which in various meta-analyses does not appear to be increased [8,9,10,11,12]. In addition, when they occur, morbidity and mortality are usually higher than in the general population [13]. Different studies have shown that both the pattern of fractures and the pathophysiological mechanisms leading to fractures seem to coincide mostly with those described in patients with obesity [13].

Although it has classically been established that obesity could be a protective factor against osteoporosis and fractures [14,15,16], in recent years there is a growing body of evidence which contradicts this. In this narrative review we will try to discuss the mechanisms by which obesity could be both a protective factor against bone loss and a risk factor for bone loss, focusing especially on the role of the chronic proinflammatory state that occurs in patients with obesity and in particular in older subjects. We will also review the role of body composition in the interpretation of diagnostic tests and fracture risk scales, osteoporosis prevention measures in this group of patients and the possible influence of obesity on the efficacy of currently available anti-osteoporotic treatments.

## 2. Epidemiological Studies: The Origin of the Paradigm Shift

The growing interest in paradigm shift on obesity as a protective factor against osteoporotic fractures stems from the results of epidemiological studies that contradict this dogma.

In the 1990s and early 2000s some studies were published which demonstrated a positive relationship between BMI and BMD [14,17,18]. Numerous studies were also published in postmenopausal women which associated a higher BMI with a lower risk of fracture, especially at the hip [19,20,21,22,23,24].

Subsequently, studies began to emerge in which the relationship between a high BMI and a lower risk of fracture did not seem so clear. In 2005, de Laet et al. [25] published a meta-analysis in which 60,000 people were included to analyze the relationship between BMI and fracture risk. As in previous studies, low BMI was associated with an increased risk. However, the linear relationship obtained disappeared when high BMIs were analyzed with respect to a normal BMI, resulting in a U-shaped relationship.

In 2009, an Italian study conducted in postmenopausal women with fracture found that a higher BMI was associated with an increased risk of humerus fracture and a lower risk of hip fracture [26]. In the same year, Beck et al. [27] used data from the Women’s Health Initiative (WHI) study in postmenopausal women and also found a lower risk of hip fracture in overweight and obese women, although the risk of lower extremity fracture was higher in this group compared to women with normal weight.

In 2010, Premaor et al. [28] looked at postmenopausal women with fractures following low-impact trauma seen over a biannual period at a Fracture Liaison Service in the United Kingdom. Of these, 28% were obese and the majority of this group had normal BMD. Compared to women without obesity, they had a significantly lower risk of wrist fracture and a significantly higher risk of hip fracture. In 2011, Compston et al. [29] conducted the multinational GLOW study, in which they found similar fracture rates in patients with and without obesity when analyzing women over 55 years of age. They also found a lower risk of wrist fracture in obese patients, although the risk of ankle and femur fracture was higher.

In 2012, Prieto-Alhambra et al. [30] conducted a prospective study in which they included more than 800,000 Spanish women over 50 years of age. As in previously commented studies, they found a lower risk of hip fracture and an increased risk of humerus fracture in patients with obesity.

Subsequently, in 2013, two Asian studies showed an increased risk of vertebral fracture associated with obesity [31,32]. In 2014, analysis of data from the Nottingham Fracture Liaison Service showed a positive relationship between ankle and shoulder fractures and obesity and a negative relationship between wrist fracture and obesity [33]. Finally, it is worth noting the meta-analysis published in 2014 [34] analyzing 25 studies and including 398,610 women with a mean age of 63 years, which showed a positive relationship between higher BMI and risk of shoulder fracture when risk was adjusted for BMD. They also found that obesity was an independent risk factor for all osteoporotic fractures.

While all these studies have been performed mainly in women, there are four relevant studies in this respect carried out in men over 65 years of age. One of them showed an association between obesity and an increased risk of non-vertebral fractures after adjustment for BMD in a cohort of 5995 patients [35]. Another study (with a cohort of more than 100,000 men) showed an increase in multiple rib fractures and a decrease in clinical vertebral fractures, hip fractures and wrist fractures in obese patients [36]. The third study included 23,061 men aged 60–79 years and the risk of fracture decreased with increasing BMI up to a plateau in obese men. On the other hand, waist circumference and waist/hip ratio were associated with an increased risk of hip fracture, especially in those with lower BMI but greater abdominal adiposity [37]. The most recent study included a sample of 1625 men over 70 years of age and showed that obesity was not a protective factor for incident fractures at 14 years regardless of whether this was classified according to BMI or body fat percentage [38]. 

On the other hand, given that there is more knowledge about the influence of body composition on health (especially cardiovascular health) beyond the simple measurement of BMI, studies that not only analyze the risk of fracture and bone fragility in relation to this parameter but also to other more specific parameters such as visceral fat or muscle mass are becoming more and more frequent. Already in 1996 Khosla et al. [14] studied the impact of muscle mass and fat mass on BMD, finding that in postmenopausal women fat mass had more influence when compared to premenopausal women. This confirmed the results of previously published studies [39,40] by Reid et al., but contradicted those published by other groups in postmenopausal women, young women and men [41,42,43]. More recent studies seem to support the findings of Khosla et al. [14,44,45], notably the study by Gnudi et al. [26] showing a relationship between BMD and both fat mass and muscle mass in women with osteoporosis but only a relationship between BMD and muscle mass in women without osteoporosis.

In relation to fat mass, several studies have analyzed whether the type of adipose tissue is also related to bone fragility. Several Asian [46,47,48] and Western [49,50] studies have shown an inverse relationship between visceral adipose tissue and BMD. Regarding fracture risk, Machado et al. [51] studied it in non-obese older women as a function of body composition, finding that visceral adipose tissue was associated with an increased risk when adjusted for other potential confounders. In 2017 Li et al. [52] conducted a meta-analysis including seven studies with a total of 551,224 patients in which they demonstrated that waist circumference and waist-to-hip ratio could be associated with increased risk of hip fracture. They concluded that indicators of abdominal obesity could be used as predictors for this type of fracture. In 2020, Gandham et al. [53] conducted a study in 1099 older subjects and showed that if patients were classified as obese based on BMI, they had a lower risk of incident fracture mediated by higher BMD; but if they were classified based on body fat percentage there was an increased risk. However, not all studies currently published have demonstrated an increase in fractures or a decrease in BMD in relation to greater visceral adipose tissue [32,54,55].

In summary, while classic epidemiological studies showed a lower risk of fractures with a higher BMI, in recent years a relationship has been shown in older subjects (especially postmenopausal women) between overweight and obesity with a higher risk of ankle, femur and humerus fracture and a lower risk of hip and wrist fracture. As for vertebral fractures, the results are more contradictory. When body composition has been analyzed in relation to bone fragility and fracture risk, it appears that muscle mass is related to BMD, while fat mass is only related to BMD in postmenopausal women. Furthermore, most studies show a negative relationship between visceral adipose tissue and BMD. Main studies focused on the relation between obesity and bone mineral density/fracture risk are shown in Table 1.

## 3. Pathophysiology

Given the results of epidemiological studies, various pathophysiological mechanisms have been investigated and described by which there could be a beneficial and/or detrimental relationship between obesity and bone fragility (Figure 1).

On the one hand, some papers have referred to the principle of Wolff’s law. According to it, bone adapts in response to the stress to which it is subjected. In the case of obesity, the mechanical overload produced would result in bone deformation that would trigger a cascade of transduction signals that would stimulate the maintenance of bone mass through osteoblastic activity and the Wnt/β-catenin signaling pathway [56]. However, this would increase the quantity but not the bone quality or bone strength, and this could explain why patients with obesity fracture with higher BMD measurements than non-obese patients [28,57]. Furthermore, if DXA results are interpreted as a function of weight, BMD results could be considered inappropriately low [58]. It is also postulated that the bone’s ability to adapt to mechanical overload is limited and is not maintained beyond a certain weight gain [57].

In relation to purely mechanical mechanisms, there also falls play an important role in the pathophysiology of fractures in patients with obesity and consequently reduced mobility. These were more frequent in women with obesity in the GLOW study and resulted in greater comorbidity [29]. In addition, falls are different from those occurring in patients without obesity, as they tend to be backward or sideways, which would explain the lower risk of wrist fracture and the higher risk of humerus fracture shown in epidemiological studies [57]. The presence of a greater amount of subcutaneous adipose tissue in the hips in gynoid-type obesities could serve as a “cushion” in falls and explain the lower risk of hip fracture shown in epidemiological studies focusing on postmenopausal women [58].

Another mechanism by which excess weight could increase BMD would be through the production of sex steroids by adipocytes, with widely known antiresorptive and anabolic effects. Regarding bone formation, estrogen increases osteogenic differentiation of mesenchymal stem cells (MSCs) and osteoblast maturation. With respect to bone resorption, estrogen inhibits osteoclast formation and induces osteoclast apoptosis [59]. Postmenopausal women with obesity and increased aromatization of androgens to estrogens in subcutaneous adipose tissue would therefore have higher levels of circulating estrogens with a positive effect on bone mass and mineralization [60]. However, it should be considered that this aromatization only occurs in subcutaneous adipose tissue, so that in obesity with a higher content of visceral adipose tissue (as occurs especially in older subjects due to the redistribution of fatty tissue related to aging) this beneficial effect is not so striking.

The same would occur in men due to the action of testosterone, which has also been associated with higher body weight [61]. However, obese men tend to have lower testosterone levels, which has also been shown to be a risk factor for falls in older men [62]. Obese men also tend to have lower levels of sex hormone binding globulin (SHBG), which leads to increased levels of free sex steroids. In fact, elevated levels of SHBG have been associated with lower BMD [63] and an increased risk of fractures [64].

Among the pathophysiological mechanisms linking obesity with increased fracture risk is 25-hydroxyvitamin D deficiency and consequent secondary hyperparathyroidism, which occur more frequently in patients with obesity. As is widely known, this has been associated with an increased risk of osteoporosis and fractures [65]. Hyperparathyroidism is also associated with increased BMD loss in cortical bone, and this may partially explain why patients with obesity have more fractures in bones such as the humerus or ankle [66]. In addition, diets rich in fat, which are sometimes part of the cause of obesity, interfere with intestinal absorption of calcium [67].

At the molecular level, several pathophysiological pathways linking obesity with an increased risk of osteoporosis have also been found. Just as the chronic proinflammatory state secondary to obesity increases cardiovascular risk, it could also increase the risk of osteoporosis and fractures by altering the mechanisms of bone formation and resorption. In fact, the proinflammatory state that occurs in other diseases such as Crohn’s disease or rheumatoid arthritis has already been widely shown to increase the prevalence of osteoporosis in those patients who present with them [68]. This pro-inflammatory state also increases the risk of insulin resistance, type 2 diabetes and arteriosclerosis, thus increasing the risk of osteoporosis and fractures as these are pathologies that also deteriorate bone quality [69]. 

Adipose tissue is currently considered as an endocrine organ, since it serves as a substrate for the synthesis of sex hormones and secretes adipokines and cytokines which, among other functions, play a role in bone metabolism [50,70,71]. Among the adipokines, leptin and adiponectin stand out. Their relationship with BMD has been widely studied with some controversy and discrepancy in the results obtained.

As for adiponectin, both osteoblasts and osteoclasts have receptors for it [72,73], and this seems to stimulate the RANKL receptor, inhibiting the production of osteoprotegerin in osteoblasts and indirectly increasing osteoclastogenesis [74]. This is why a negative relationship between adiponectin levels and BMD has been found even after adjusting these results for total fat mass for very different groups of age and BMI [75,76,77,78]. However, this relationship in most studies is not confirmed when only premenopausal women are analyzed [79], so it is believed that the levels of sex hormones could be a confounding factor which should be studied. On the other hand, the adiponectin effects could also be mediated by its influence on insulin levels. People with obesity have decreased levels of adiponectin compared to normal weight people, especially in those with type 2 diabetes, insulin resistance and central obesity, and it is postulated that this could be a mechanism by which obesity could be a protective factor for osteoporosis and fractures [80]. 

In contrast, leptin, which is increased in patients with obesity, has receptors on osteoblasts and appears to directly stimulate osteoblast cell differentiation and inhibit osteoclast cell differentiation [81]. This could be mediated by its inhibition of the activated NF-kB ligand receptor and a consequent increased expression of osteoprotegerin [82]. However, it also activates the sympathetic nervous system at the hypothalamic level, which would inhibit bone formation [83]. Currently, the relationship between leptin levels and BMD in humans shows contradictory results and it is not clear whether lectin levels ultimately have a beneficial, detrimental or neutral role in bone tissue or whether their effect is a reflection of the percentage of total fat mass [71,78,81,84]. In one study, a positive relationship between leptin levels and BMD was found to be greater in menopausal women with obesity than in the rest of the studied patients [85], which could be due to a state of resistance to leptin in the central nervous system in patients with obesity [86]. 

Moreover, cytokines produced in adipose tissue play a crucial role in the relationship between obesity, osteoporosis and fractures. Patients with obesity produce higher levels of cytokines such as IL-6, C-reactive protein and TNF-α than normoweight individuals [87,88]. Osteoblasts and adipocytes derive from common precursor stem cells [89]. The differentiation of these cells into adipocyte or osteoblast depends on the activation of a series of cytokines (PPAR-α and CEBP-α, β and Δ in the case of adipocyte and RUNX2, BMP2 and TGF-β among others in the case of osteoblast) [89]. This could even be reversed and the adipocyte returned to the precursor cell state to finally differentiate into osteoblast [90]. It is therefore evident that there is a strong relationship between adipose and bone tissue, which depends on a cytokine microenvironment that can be altered in proinflammatory states. In patients with obesity, PPAR-α levels are increased in adipose tissue [91], and in animal models this has also been related to fat distribution [92].

In addition to influencing stem cell differentiation, the proinflammatory state that occurs in obesity increases the levels of cytokines that stimulate osteoclast formation and activity by affecting the RANKL/RANK/OPG pathway, such as TNF-α and IL-6 [93,94]. In particular, TNF-α has a direct and indirect pro-osteoclastogenic effect, as it promotes RANKL expression in bone marrow stromal cells [95]. Obesity has also been associated with increased secretion of RANKL by osteoblasts [96]. As for osteoblastogenesis, it is inhibited by proinflammatory cytokines and other substances are increased in obese patients such as advanced glycosylation products (AGEs) and sclerostin [97,98]. 

Another cytokine increased in patients with obesity and related to bone tissue is Monocyte Chemotactic Protein-1 (MCP1). This is expressed by various normal cells, such as fibroblasts [99], smooth muscle cells [100], mesotelial cells [101], adipocytes [99], chondrocytes [102] and osteoblasts [103] among others. The expression of this cytokine is increased in tumor cells [104] and may also be downregulated when receiving corticosteroid treatment and with the increase of nitric oxide and other cytokines such as IL-13 [105,106,107]. The number of MCP1 receptors as well as their levels are increased in the visceral and subcutaneous adipose tissue of obese patients when compared to patients without obesity [108]. This is relevant since MCP1 has a pro-angiogenic action and contributes to adipose tissue expansion [109]. In addition, it interacts with the CCR2 receptor present in monocytes and macrophages and this stimulates osteoclastogenesis through the JAK/STST and Ras/MPAK pathways [110].

On the other hand, TRAIL is a cytokine belonging to the TNF superfamily. In undifferentiated osteoblasts it induces a pro-apoptotic signal [111] and directly induces osteoclastogenesis in the absence of RANKL, while in its presence it has an inhibitory action [112]. As for adipose tissue, it determines an inflammatory state [113] and induces the proliferation of preadipocytes [114], forming part of the pathogenesis of obesity and other metabolic diseases [115].

Osteoprotegerin (OPG) is a soluble TRAIL with anti-inflammatory and anti-apoptotic effects [116]. Its levels are reduced in states of obesity [117,118], insulin resistance [117] and some of its complications such as non-alcoholic fatty liver disease [119], although this has not been confirmed in all studies [120].

The receptor LIGHT (a cellular ligand for herpes virus entry mediator and lymphotoxin receptor) is expressed by T-lymphocytes and it also belongs to the TNF superfamily. It is increased in patients with obesity and has a pro-osteoclastogenic effect. It has been shown that an elevation of its levels is related to osteoporosis [121,122].

The production of proinflammatory cytokines is higher in abdominal fat, while adiponectin secretion and aromatase activity is lower than in subcutaneous adipose tissue [123,124]. In addition, increased levels of these cytokines decrease adiponectin production. Since adiponectin could have a beneficial effect on BMD, this decrease in adiponectin levels would be detrimental [125].

Other markers of inflammation such as C-reactive protein are also increased in patients with obesity (especially in those with abdominal obesity) [67]. This has been associated with decreased levels of bone remodeling markers as well as lower BMD.

Finally, the role of some hormones in bone metabolism is also interesting, especially insulin. Its levels are usually elevated in patients with obesity due to a state of insulin resistance. Insulin is an anabolic hormone that contributes to bone formation by directly stimulating osteoblasts [15]. It also reduces hepatic production of SHBG, increasing the bioavailability of estrogens and androgens [13]. However, in states of insulin resistance, it has a direct effect on osteoclastic cells by reducing the carboxylation of osteocalcin, essential for bone mineralization. In addition, it increases the production of RANKL, increasing bone resorption [126,127]. Thus, insulin resistance negatively affects bone tissue [15,128]. However, it is controversial whether this is a direct effect or a reflection of the effects of other factors usually associated with insulin-resistant states [129].

The role of ghrelin, a hormone increased in patients with obesity, is also still very controversial in bone metabolism and requires further study. Although in vitro a protective role on bone tissue seems to have been demonstrated [130], human studies have shown an association only with trabecular BMD [131]. In studies performed in bariatric surgery, the reduction of ghrelin after this procedure was associated with a greater loss of BMD [132].

## 4. Changes in Body Composition during Aging

Aging produces various changes in body composition independently of changes in weight.

In terms of muscle mass, between the ages of 24 and 50 years, 10% of muscle mass is lost, to which is added a 30% loss between the ages of 50 and 80 years, with a 1% annual decrease in the fifth decade of life [133]. This can lead to a state of sarcopenia, which is a state of decreased muscle mass and strength associated with functional limitations that may increase the risk of falls [134]. The prevalence of sarcopenia is estimated to be 5–13% in patients aged 60–70 years, increasing to 50% in patients aged 80 years or older and being more prevalent in patients with metabolic and chronic diseases [135]. One of the main characteristics of sarcopenia in the older subjects is fatty infiltration of muscle [136,137], which has been associated with an increased risk of fractures [138].

Obesity, due to the related chronic proinflammatory state, may contribute to a greater development of sarcopenia than that produced by aging itself [139]. The presence of obesity in patients with sarcopenia is referred to as sarcopenic obesity, which has been associated with an increased risk of morbidity and mortality [140]. Given the aging population and the increasing prevalence of obesity, this combination is becoming increasingly prevalent leading to a public health problem, especially given the resulting increase in cardiovascular risk [141]. It is estimated that the most important cause of fatty deposits in skeletal muscle is due to energy intake exceeding energy expenditure, resulting in energy storage in the form of adipose tissue. In people with obesity this is increased, since they have enlarged adipocytes in the subcutaneous tissue and an overload of lipid deposits that cause this excess fat to accumulate in other tissues such as muscle, following the “overflow hypothesis” [142]. In addition to this lipid overload, adipocytes in people with obesity have a lower capacity for lipid accumulation than adipocytes in people without obesity. This fact is due to the proinflammatory state of obesity, since the increased levels of IL-6 and TNF-α reduce the expression of PPAR-γ-2 and C/EBPα, which play an important role in the correct differentiation of preadipocytes into adipocytes [143].

On the other hand, the distribution of fat tissue itself also changes with aging, increasing visceral adipose tissue and decreasing subcutaneous adipose tissue, which goes on to infiltrate other organs such as muscle [144]. In particular, fatty infiltration of the bone marrow is relevant, which has been related to lower bone quality [145]. As we age, the capacity of preadipocytes to replicate, differentiate and resist apoptosis decreases due to the increase in inflammation parameters. This phenomenon is known as inflammaging [146]. As we have mentioned, this redistribution is accentuated in patients with obesity also due to alteration of the regulatory mechanisms of inflammation. This change, especially due to the increase in visceral fat, increases cardiovascular risk in older subjects, with greater relevance in patients who are also obese. As previously described, the increase in visceral adipose tissue has also been associated with a decrease in BMD and an increased risk of fractures.

Finally, it has been shown that older subjects may be particularly susceptible to the deleterious effects of obesity, since the correlation between BMI and frailty is U-shaped in these subjects, presenting the obese older subjects less aerobic capacity, less muscle strength, less physical performance and worse functionality [147].

In summary, aging produces changes in body composition (redistribution of adipose tissue with a decrease in subcutaneous fat and an increase in visceral, intramuscular—with the consequent sarcopenia—and bone marrow deposits) that are associated with greater bone fragility and an increased risk of falls and fractures. This redistribution is similar to that produced in obesity and therefore its deleterious effects are increased in older subjects and obese patients.

## 5. Difficulties in the Diagnosis of Osteoporosis and Prediction of Fracture Risk in Patients with Obesity

As previously discussed, obese patients show higher BMD compared to patients without obesity on DXA. However, higher BMI and greater soft tissue thickness could alter this measurement [148]. In addition, it seems that BMD assessment by DXA may provide inappropriate values if not interpreted in relation to weight.

As for other tests less widely used in daily clinical practice, such as high-resolution peripheral quantitative computed tomography (HRpQCT), a greater BMD has also been shown in patients with obesity, as well as greater cortical and trabecular BMD and a greater number of trabeculae in the distal radius and distal tibia, where they also present greater bone strength [149,150]. However, the bone size in the tibia and radius measured by this technique is not increased with respect to patients with normal weight, unlike the hip area [149,151]. This is in contradiction with the theory that mechanical overload in patients with obesity would contribute to increased bone formation. This technique also allows the calculation of the amount of adipose tissue in the bone marrow, which is usually increased in patients with obesity and in the older subjects and which has been related to bone microstructural deterioration and the presence of non-vertebral fractures [152]. Like DXA, the accuracy of this test is also influenced by the thickness of the soft tissue [153].

In patients with type 2 diabetes, a cortical strength deficit has been observed by HRpQCT, due to reduced cortical thickness and volume with increased cortical porosity in patients with microvascular complications [154]. This has also been found to be increased in patients with type 2 diabetes with previous fractures, so it seems that these changes would contribute to an increased risk of fractures in these patients [155].

As for bone remodeling markers, these are found to be decreased in patients with obesity when compared to patients with normal weight, this difference being greater in bone resorption markers than in bone formation markers [156]. This reduction has also been demonstrated in patients with type 2 diabetes, independently of glucose levels [157], which is in agreement with the results of histomorphometric studies in which signs compatible with low bone remodeling are observed [158].

Regarding fracture risk, tools such as FRAX can underestimate it in these patients. As we know, given the description of the increase in fractures in relation to BMIs below normal, this is a parameter that is considered in this algorithm. However, given the results of older epidemiological studies previously discussed, obesity is not included as a risk factor for fractures in this tool.

There are studies that have evaluated the sensitivity of FRAX in this group of patients. In 2013, Premaor et al. [159] compared obese postmenopausal women with non-obese women, observing that the probability calculated by FRAX for fracture at 10 years was significantly lower in the first group (7.1% vs. 10.9% in hip fracture and 18.2% vs. 23.3% in major osteoporotic fracture respectively), even if BMI was not included in the calculation (5.8% vs. 11.4% in hip fracture and 17.6% vs. 23.6% in major osteoporotic fracture). Despite this, when calculating the ROC curve, the area under the curve was similar in both groups with and without the inclusion of BMI in the calculation. It therefore suggests that the cut-off values at which to intervene may be too high for patients with obesity and lower reference values should be considered for initiating treatment. Moreover, the percentages of predicted and subsequently observed fractures were similar between groups. Another study conducted in 2014 [160] also showed that the ability of FRAX to predict fractures did not vary with body composition.

However, the FRAX tool has two important limitations in patients with obesity: the first is that it does not predict fractures that are more frequent in this group of patients, such as ankle fractures; the second is that in patients with type 2 diabetes, increased waist circumference and/or insulin resistance it has been shown to underestimate the risk of fracture [161]. Considering the current prevalence of obesity in older subjects, more studies are needed in the coming years to clarify this issue because of its implications.

It should be noted that, as mentioned above, patients with obesity who undergo an osteoporosis study should also be asked to have their HbA1c level measured, since diabetes influences the interpretation of the tests.

## 6. Prevention of Osteoporosis and Fractures in Older Subjects with Obesity

For the prevention of osteoporosis and fractures in patients with obesity, special emphasis should be placed on lifestyle measures. As in the general population, smoking and alcohol intake cessation should be advised, as well as physical exercise and a healthy diet.

Weight loss has been associated with a 1–4% loss of bone mass in the hip and trabecular bones [162,163,164,165], especially in older subjects [166,167]. When it occurs involuntarily, it has been associated with an increase in hip and upper limb fractures [168], but this may be due to the loss of muscle mass that occurs when weight is lost involuntarily rather than the weight reduction itself. Studies that have evaluated intentional weight loss have shown increases in lower leg risk but a decrease in hip, pelvic and spine fractures [168]. That distribution of fractures is similar to that occurring in patients with obesity, who are the most likely to undertake an intentional weight loss program, so these results could be biased. Furthermore, recent studies have shown that when weight loss is moderate, BMD is not reduced and bone geometry is not altered [169]. Compared to this moderate weight loss, intense caloric restriction in a randomized clinical trial resulted in a greater loss of BMD in the hip in postmenopausal women, but not in the lumbar spine [170]. In this same group of patients, a study showed that when BMD is lost after weight loss, it does not recover if the lost weight is regained [171].

Given the relevance of sarcopenia in obese older patients with respect to the risk of osteoporosis and fractures, multiple studies have evaluated the role of physical exercise in these weight loss programs. In older obese patients undergoing a weight loss program, physical exercise has been shown to reduce frailty and decrease BMD and sarcopenia [147,172,173] with both resistance exercise programs [173] and aerobic combined with resistance exercise programs [172]. On the other hand, dairy intake during weight loss has been associated with higher osteocalcin levels and increased BMD in the lumbar spine when compared to low dairy intake [174]. In summary, in older patients with obesity, moderate weight loss should be advised in a program that includes adequate dairy intake and resistance exercise.

Regarding diet, it has been described, in experimental models, that hypercaloric and obesogenic diets are related to an increased risk of fracture by direct and indirect mechanisms [175,176]. High-fat diets are a risk factor for osteoporosis. In mice subjected to this type of diet, T lymphocytes isolated from the spleen and bone marrow showed increased expression of RANKL, and these mice had decreased BMD [177], as well as increased levels of cytokines such as IL-6 and TNF-α. It has also been shown in animal models that this type of diet affects bone remodeling, triggering a loss of trabecular bone mass and also reduces calcium absorption [178]. On the other hand, a high-fat and high-sucrose diet has been shown to affect the cortical bone in mice and rats, especially when maintained over the long term [179,180,181]. In humans, data regarding the effect of a high-fat diet on the risk of osteoporosis and fracture are scarce and contradictory [182]. However, some prospective and cross-sectional studies have shown a protective effect with protein intake [182].

It is also worth noting the importance in these patients of an adequate intake of calcium and vitamin D, since as indicated above, high-fat diets tend to decrease calcium absorption and these patients have a high prevalence of vitamin D deficiency. As in the general population, it is recommended to obtain an optimal calcium and vitamin D intake through diet and not with supplementation if possible, especially with regard to calcium supplements that could increase arteriosclerosis [183]. As for vitamin D, given its accumulation in adipose tissue, higher doses are usually required than in the general population.

Current measures to reduce osteoporosis and fracture risk in obesity are shown in Figure 2.

## 7. Effectiveness of Osteoporotic Treatments in Obesity

The current evidence on the effectiveness of osteoporotic treatments in the prevention of fractures is based on studies carried out in postmenopausal women with low BMD with or without fractures, making it difficult to extrapolate these results to older men and women with obesity and BMD which is not so low. Contributing to this is the fact that most of the drugs used for osteoporosis (and the only ones whose effectiveness has been studied in patients with obesity at present) are antiresorptive treatments, while patients with obesity tend to have both bone remodeling and bone resorption already reduced.

In this regard, a randomized clinical trial in which clodronate was used in women stands out. It showed a reduction in osteoporotic fractures in non-obese women, but not in obese women [184]. On the other hand, a subanalysis of the FREEDOM study showed a similar vertebral fracture reduction in obese and non-obese postmenopausal women with denosumab, although this significant reduction in non-vertebral fractures was only obtained in those without obesity [185]. The same occurred in the subanalysis of this same study comparing patients with and without diabetes [186]. In the HORIZON study, treatment with annual intravenous zoledronic acid had a greater decrease in the risk of vertebral fracture in patients with BMI ≥ 25 kg/m^2^, but not in non-vertebral fractures [187]. These data have led some authors to consider whether the dose of antiresorptive drugs should be increased in patients with obesity to also reduce the risk of non-vertebral fracture [187,188].

Finally, in the GLOW study [29], it was observed that obese women with fractures were treated in 27% compared to 41% of non-obese women. This may be because they often have fractures not considered as osteoporotic (e.g., in the ankle) and because they have higher BMD than patients who are normal or underweight.

Regarding osteoanabolic treatments, teriparatide and anti-sclerostin antibodies have demonstrated an increase in BMD in rats with diabetes, but their phenotype is different from that of humans with type 2 diabetes, so the results cannot be extrapolated and more research should be carried out in this regard [189,190].

## 8. Methods

A search of the scientific literature published in PubMed through March 2022 was conducted to identify peer-reviewed articles on obesity, corporal composition, healthy aging, osteoporosis and fracture risk.

Three computerized electronic databases (PubMed, Web of Science and Scopus) were searched using the following key search words: (“high-fat diet” OR “exercise programs” OR “cytokines” OR “sarcopenic” OR “FRAX” OR “BMD” OR “adipocyte” OR “inflammaging” OR “insulin resistance” OR “body composition”) AND (“osteoporosis” OR “fracture”) AND (“elderly” OR “postmenopausal”) AND (“obesity”)

Original human and animal models research articles published in English, prospective and retrospective observational studies, randomized controlled trials, editorials, opinions and letters to the editor were included. The largest studies and the most recent and strongest available evidence were prioritized. All possible articles were merged into a single file, and duplicate records were removed after they were checked manually.

## 9. Conclusions

Despite the classic concept of obesity as a protective factor for fractures, the most recent evidence has shown that these patients, especially those with greater visceral adipose tissue and less muscle mass (changes that occur with aging), present an increased risk of incident fractures, which represents a paradigm shift. Among the pathophysiological mechanisms, the chronic proinflammatory state that occurs in these patients, widely investigated in relation to cardiovascular risk but not so much in relation to bone metabolism, stands out. Although weight loss has been associated with BMD losses, when this is carried out in the context of physical exercise programs and the abandonment of high-fat diets, this effect seems to disappear. Therefore, in older subjects with obesity, these measures could be recommended to reduce the risk of fracture. More research is needed on the usefulness and adequacy of diagnostic tests and fracture risk scales in this group of patients, as well as on the efficacy of currently available antiosteoporotic treatments.

It is important for professionals to be aware of the increased risk of osteoporosis and fracture in patients with obesity, as they are currently considered low risk and tend to be underdiagnosed and undertreated. This may have a major impact on the health of this group of patients and also socioeconomic consequences, especially with the increase in both pathologies expected in the coming years.

## Figures and Tables

**Figure 1 ijms-23-08303-f001:**
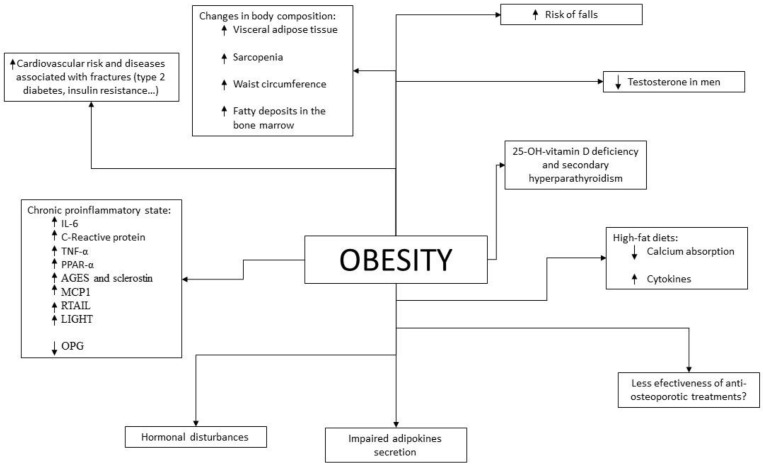
Pathophysiological mechanisms that relate obesity to bone health. Arrows indicate increase or decrease levels.

**Figure 2 ijms-23-08303-f002:**
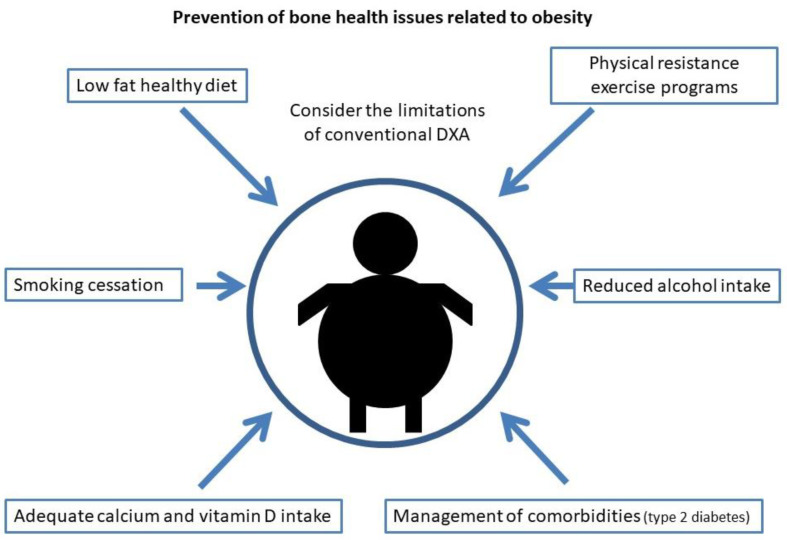
Current measures for the prevention of bone health issues related to obesity.

**Table 1 ijms-23-08303-t001:** Main studies focused on the relation between obesity and bone mineral density/fracture risk.

Author, Year	Country	Type of Study	Subjects	Statistical Results	Results
Felson, 1993 [17]	U.S.	Cohort study	1132 older male and female subjects of the Framingham osteoporosis study	After adjusting for other factors affecting bone density, both recent weight and BMI explained a substantial proportion of the variance in BMD for all sites in women (8.9–19.8% of total variance, all *p* < 0.01) and for only weight-bearing sites (femur and spine) in men (2.8–6.9% of total variance, all *p* < 0.01). For BMD at the proximal radius, weight and BMI accounted for <1% of variance in men.	There was a positive relation between BMI and BMD. After adjusting for other factors affecting bone density, both recent weight and BMI explained a substantial proportion of the variance in BMD for all sites in women and for only weight-bearing sites in men.
Joakimsen RM, 1998 [22]	Norway	Prospective population-based study	12,270 (922 persons with fractures) middle-aged	Change in BMI was not associated with fractures among men, except for a lower incidence of hip fractures (not only low-energy) among those who had gained weight (RR 0.69, 95% CI 0.50–0.95, age adjusted per unit BMI increase). Women who had an increase in their BMI had a lower risk of all low-energy fractures (RR 0.95, 95% CI 0.90–1.01, age adjusted per unit BMI increase) and of low-energy fractures in the lower extremities (RR 0.88, 95% CI 0.80–0.97, age adjusted per unit BMI increase)	The risk of a low-energy fracture was found to be positively associated with increasing body height and with decreasing BMI. High body height was a risk factor for fractures, and 1 in 4 low-energy fractures among women today might be ascribed to the increase in average stature since the turn of the century
De Laet, 2005 [25]	Multinational	Meta-analysis	Almost 60,000 men and women from 12 prospective population-based cohorts, with a total follow-up of over 250,000 subjects.	The RR per unit higher BMI was 0.98 (95% confidence interval [CI], 0.97–0.99) for any fracture, 0.97 (95% CI, 0.96–0.98) for osteoporotic fracture and 0.93 (95% CI, 0.91–0.94) for hip fracture (all *p* < 0.001). The RR per unit change in BMI was very similar in men and women (*p* > 0.30). After adjusting for BMD, these RR became 1 for any fracture or osteoporotic fracture and 0.98 for hip fracture (significant in women). A BMI of 30 kg/m^2^, when compared with low BMI, confers a risk of substantial importance for all fractures that is largely independent of age and sex, but dependent on BMD. The significance of BMI as a risk factor varies according to the level of BMI of 25 kg/m^2^, was associated with only a 17% reduction in hip fracture risk (RR = 0.83; 95% CI, 0.69–0.99)	Low BMI confers a risk of substantial importance for all fractures that is largely independent of age and sex, but dependent on BMD. The significance of BMI as a risk factor varies according to the level of BMI.
Gnudi, 2009 [26]	Italy	Cross-sectional study	2235 postmenopausal women with fragility fractures (hip, ankle, wrist and humerus)	BMI had a protective effect against hip fracture: OR 0.949 (0.9–0.999) and higher risk of humerus fracture: OR 1.077 (1.017–1.141)	Risk of hip fracture increases as BMI decreases. The risk of humerus fractures increases as BMI increases.
Beck, 2009 [27]	US	Cohort study	A subset of 4642 postmenopausal non-Hispanic whites (NHWs) from the Women’s Health Initiative Observational Cohort (WHI-OS). Age 59–70 years old.	Femur BMD in overweight: 0.706 (*p* = 0.002 when compared to healthy weight). Rates of central body fractures decline significantly with BMI and were 40% less likely in the extremely obese	Femur BMD and geometric strength are greater with overweight in post-menopausal women, but they vary proportion to lean (mostly muscle) mass and not to body weight or fat mass. Femur strength is reduced relative to body weight in the obese but although obese women reported more falls they had fewer fractures at hip and other central body sites.
Premaor, 2010 [28]	UK	Cohort study	805 postmenopausal women aged less than 75 years with a low-trauma fracture.	Normal BMD was reported in 59.1% of obese and 73.1% of morbidly obese women, and only 11.7% and 4.5%, respectively, had osteoporosis (*p* < 0.001). A significant positive association with BMI (*p* < 0.001) and previous fracture (*p* < 0.001) was found.	There was a high prevalence of obesity in postmenopausal women presenting with low-trauma fracture. Most of these women had normal BMD, as measured by DX. A higher BMI was associated with a higher rate of previous fracture.
Compston, 2011 [29]	Multinational	Prospective observational population-based study	60,393 women aged ≥ 55 years	Fracture prevalence in obese women at baseline was 222 per 1000 and incidence at 2 years was 61.7 per 1000, similar to rates in nonobese women (227 and 66.0 per 1000, respectively). The risk of incident ankle (adjusted odds ratio [OR] 1.5; 95% confidence interval [CI], 1.2–1.9) and upper leg (OR 1.7; 95% CI, 1.1–2.5) fractures was significantly higher in obese than in nonobese women, while the risk of wrist fracture was significantly lower (OR 0.8; 95% CI, 0.6–1.0).	Obesity is not protective against fracture in postmenopausal women and is associated with increased risk of ankle and upper leg fractures.
Prieto-Alhambra, 2012 [30]	Spain	Cross-sectional study	832775 women aged ≥ 50 years.	Hip fractures were significantly less common in overweight and obese women than in normal/underweight women (RR 0.77 (95% CI 0.68 to 0.88), RR 0.63 (95% CI 0.64–0.79), *p* < 0.001 respectively). Pelvis fracture rates were lower in the overweight (RR 0.78 (95% CI 0.63–0.96), *p* = 0.017) and obese (RR 0.58 (95% CI 0.47–0.73), *p* < 0.001) groups. Conversely, obese women were at significantly higher risk of proximal humerus fracture than the normal/underweight group (RR 1.28 (95% CI 1.04–1.58), *p* = 0.018)	Obese women with hip, clinical spine and pelvis fracture were significantly younger at the time of fracture than normal/underweight women, whereas those with wrist fracture were significantly older. The association between obesity and fracture in postmenopausal women is site-dependent, obesity being protective against hip and pelvis fractures but associated with an almost 30% increase in risk of proximal humerus fractures when compared with normal/underweight women.
Tanaka, 2013 [31]	Japan	Cohort study	1614 postmenopausal Japanese women	Incidence rates of vertebral fracture in underweight and normal weight women were significantly lower than overweight or obese women by 0.45 (95% CI: 0.32 to 0.63) and 0.61 (0.50 to 0.74), respectively, if BMD and other risk factors were adjusted, and by 0.66 (0.48 to 0.90) and 0.70 (0.58 to 0.84) if only BMD was not adjusted. Incidence rates of femoral neck and long-bone fractures in the underweight group were higher than the overweight/obese group by 2.15 (0.73 to 6.34) and 1.51 (0.82 to 2.77) and were similar between normal weight and overweight/obesity.	Overweight/obesity and underweight are both risk factors for fractures at different sites. Vertebral fracture was more frequent in overweight and obese women and femoral neck and long bones fractures were less frequent in these groups when compared to underweight/normal weight groups.
Ong, 2014 [33]	UK	Cross-sectional study	4288 women and men >50 years old, with a low trauma fracture from 1 January to 31 August 2007. Data were collected from the Nottingham Fracture Liaison Service.	Prevalence of osteoporosis was 13.4%, 24.9% and 40.4% in the obese, overweight and normal category respectively. Being obese has an odds ratio of 0.23 (95% CI 0.19–0.28, *p* < 0.01) of having osteoporosis compared to a normal BMI category. Obese patients, when compared with the non-obese category, were more likely to fracture their ankle (OR 1.48, *p* < 0.01) and upper arm (OR 1.48, *p* < 0.001), but were less likely to fracture their wrist (OR 0.65, *p* < 0.001). In the older subjects (>70 years), obesity no longer influenced ankle or wrist fractures but there is an increased risk of upper arm fractures (OR 1.46, *p* = 0.005).	Higher BMD in obesity is not protective against fractures. Despite obese people having less osteoporosis, they are more likely to present with ankle and upper arm fractures and less likely to present with wrist fracture.
Kaze, 2014 [34]	Multinational (countries from Europe, Asia, North America)	Meta-analysis	105,129 participants followed for 3 to 19 years.	The pooled RR (95% CI for vertebral fracture) per each standard deviation increase in BMI was 0.94 (95% CI = 0.80–1.10) with significant heterogeneity (I^2^ = 88.0%, *p* < 0.001). In subgroup analysis by gender, a significant inverse association between BMI and risk of vertebral fracture in men (RR = 0.85, 95% CI = 0.73–0.98, *n* = 25,617 participants) was found, but not in women (RR = 0.98, 95% CI = 0.81–1.20, *n* = 79,512 participants). Across studies of women not adjusting for BMD, there was no significant association between BMI and risk of vertebral fracture (RR = 0.91, 95% CI = 0.80–1.04, *p* = 0.18, *n* = 72,755 participants). BMI was associated with an increased risk of vertebral fracture in studies of women adjusted for BMD (RR = 1.28, 95% CI = 1.17–1.40, *p* < 0.001, *n* = 6757 participants). Substantial heterogeneity was found among studies of women (I^2^= 90.1%, *p* < 0.001).	There are gender differences in the relationship of BMI with risk of vertebral fracture. BMI was associated with an increased risk of vertebral fracture in studies of women that adjusted for BMD.
Nielson, 2011 [35]	US	Cohort study	5995 men 65 years of age and older.	In age-, race-, and BMD-adjusted models, compared with normal weight, the hazard ratio (HR) for non-spine fracture was 1.04 [95% CI 0.87–1.25] for overweight, 1.29 (95% CI 1.00–1.67) for obese I, and 1.94 (95% CI 1.25–3.02) for obese II. Associations were weaker and not statistically significant after adjustment for mobility limitations and walking pace (HR = 1.02, 95% CI 0.84–1.23, for overweight; HR = 1.12, 95% CI 0.86–1.46, for obese I, and HR = 1.44, 95% CI 0.90–2.28, for obese II).	When BMD is held constant, obesity is associated with an increased risk of non-spine fracture in older male subjects.
Premaor, 2013 [36]	Spain	Population-based cohort study	139,419 men ≥65 years. Men were categorized as underweight/normal (BMI < 25 kg/m^2^, *n* = 26,298), overweight (25 ≤ BMI < 30 kg/m^2^, *n* = 70,851) and obese (BMI ≥ 30 kg/m^2^, *n* = 42,270).	A statistically significant reduction in clinical spine and hip fractures was observed in obese (RR, 0.65; 95% CI, 0.53–0.80 and RR, 0.63; 95% CI, 0.54–0.74, respectively) and overweight men (RR, 0.77; 95% CI, 0.64–0.92 and RR, 0.63; 95% CI 0.55–0.72, respectively) when compared with underweight/normal men. Additionally, obese men had significantly fewer wrist/forearm (RR, 0.77; 95% CI, 0.61–0.97) and pelvic (RR, 0.44; 95% CI, 0.28–0.70) fractures than underweight/normal men. Conversely, multiple rib fractures were more frequent in overweight (RR, 3.42; 95% CI, 1.03–11.37) and obese (RR, 3.96; 95% CI, 1.16–13.52) men.	In older men, obesity is associated with a reduced risk of clinical spine, hip, pelvis, and wrist/forearm fracture and increased risk of multiple rib fractures when compared to normal or underweight men.
Li, 2017 [52]	Multinational (Europe, North America	Meta-analysis	Seven studies involving 180,600 participants for hip circumference, six studies involving 199,828 participants for waist–hip ratio and five studies involving 170,796 participants for waist circumference were included.	The combined RRs with 95% CIs of hip fracture for the highest versus lowest category of waist circumference, waist–hip ratio, and hip circumference were 1.58 (95% CI 1.20–2.08), 1.32 (95% CI 1.15–1.52) and 0.87 (95% CI 0.74–1.02), respectively. For dose-response analysis, a nonlinear relationship was found (P_nonlinearity_ < 0.001) between waist circumference and the risk of hip fracture, and a linear relationship (P_nonlinearity_ = 0.911) suggested that the risk of hip fracture increased about 3.0% (1.03 (1.01–1.04) for each 0.1 unit increment of waist–hip ratio.	Abdominal obesity as measured by waist circumference and waist–hip ratio might be associated with an increased risk of hip fracture.
Gandham, 2020 [53]		Cohort study	1099 older subjects. Obesity status at baseline was defined by BMI (≥30 kg/m^2^) obtained by anthropometry and body fat percentage (≥30% for men and ≥40% for women) assessed by dual-energy X-ray absorptiometry (DXA).	Prevalence of obesity was 28% according to BMI and 43% according to body fat percentage. Obese older subjects by BMI, but not body fat percentage, had significantly higher aBMD at the total hip and spine compared with non-obese (both *p*-value < 0.05). Obese older subjects by body fat percentage had significantly higher likelihood of all incident fractures (OR: 1.71; CI:1.08, 2.71) and non-vertebral fractures (OR: 1.88; CI:1.16, 3.04) compared with non-obese after adjusting for confounders. Conversely, obese older subjects by BMI had a significantly lower likelihood (OR: 0.54; CI:0.31, 0.94) of non-vertebral fractures although this was no longer significant after adjustment for total hip aBMD (all *p*-value > 0.05).	Obesity defined by body fat percentage is associated with increased likelihood of incident fractures in community-dwelling older subjects, whereas those who are obese according to BMI have reduced likelihood of incident fracture.

## Data Availability

Not applicable.

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
