# Peer review of "Obesity and Bone Health: A Complex Relationship"

_ijms, 2022, doi:10.3390/ijms23158303_

Round 1

Reviewer 1 Report

The present submission is interesting. A few concerns should be addressed. 

(1) The first 3 paragraphs may be merged into one. 

(2) Line 85, there are non space between the words. Please eliminate the similar mistakes throughout the text.

(3) The phrases in the Table 1 is not correctly shown, some of words lacks space in between.

(4) Line 315 to 318 is a little confusing.

(5)Please check the meaning of the Line 323.

(6) A figure to illustrate the current measure to reduce this secondary osteoporosis is needed.

(7) There are grammatical mistakes in the text.

(8) The conclusion should be more inspiring. 

Author Response

The present submission is interesting. A few concerns should be addressed. 

First, we want to thank for your effort in reviewing our manuscript and for your constructive comments, which have undoubtedly contributed to improve the quality of our manuscript. Please, find below the responses to your kind suggestions:

1. The first 3 paragraphs may be merged into one. 

According to your suggestion, the 3 first introduction paragraphs have been merged in the revised version of the manuscript.

2. Line 85, there are non space between the words. Please eliminate the similar mistakes throughout the text.

Thank you for your advice. Accordingly, we have checked and corrected these typographical errors through the manuscript.

3. The phrases in the Table 1 is not correctly shown, some of words lacks space in between.

Thank you for your comment. These errors have been revised and corrected in the new version of the manuscript.

4. Line 315 to 318 is a little confusing.

According to your suggestion we have modified this sentence to make it clearer as follows: The receptor LIGHT (a cellular ligand for herpes virus entry mediator and lymphotoxin receptor) is expressed by T-lymphocytes and it also belongs to the TNF superfamily”

5. Please check the meaning of the Line 323.

We have divided the sentence in two for easier reading as follows: “In addition, increased levels of these cytokines decrease adiponectin production. Since adiponectin could have a beneficial effect on BMD, this decrease in adiponectin levels would be detrimental”.

6. A figure to illustrate the current measure to reduce this secondary osteoporosis is needed.

Thank you for your recommendation. Accordingly, a new illustrative figure about this issue has been added in the revised version of the manuscript.

7. There are grammatical mistakes in the text.

Thank you for your advice. Based on this, we have revised and corrected all the mistakes we have detected in the new version of the manuscript.

8. The conclusion should be more inspiring. 

We are very grateful for your recommendation. We have modified some parts of the conclusion section in order to emphasize the importance of the awareness of professionals in this area as follows:

“Despite the classic concept of obesity as a protective factor for fractures, the most recent evidence has shown that these patients, especially those with greater visceral adipose tissue and less muscle mass (changes that occur with aging), present an increased risk of incident fractures, which represents a paradigm shift”…

… “It is important for professionals to be aware of the increased risk of osteoporosis and fracture in patients with obesity, as they are currently considered low risk and tend to be underdiagnosed and undertreated. This may have a major impact on the health of this group of patients and also socioeconomic consequences, especially with the increase in both pathologies expected in the coming years.

Reviewer 2 Report

The manuscript a narrative review on potential impact of obesity on fracture and bone health. The topic is interesting and relevant, and manuscript is easy to understand available, pertinent literature on the topic. I have only few comments.

1. Please specify that the review is a 'narrative review'.  Also, in the 'methods' section, please provide search strategy for references. 

2. Term 'elderly can be replaced with more inclusive words - check the recently published paper in JAGS : https://www.ncbi.nlm.nih.gov/pmc/articles/PMC6379683/

3. There are a few sentences where editing should be well performed -- an example is " peoplewereincludedtoanalyzetherelationshipbetween 85"

Author Response

The manuscript a narrative review on potential impact of obesity on fracture and bone health. The topic is interesting and relevant, and manuscript is easy to understand available, pertinent literature on the topic. I have only few comments.

First, we want to thank for your effort in reviewing our manuscript and for your constructive comments, which have undoubtedly contributed to improve the quality of our manuscript. Please, find below the responses to your kind suggestions:

1. Please specify that the review is a 'narrative review'.  Also, in the 'methods' section, please provide search strategy for references. 

Thank you for your comment. Accordingly, we have specified that it is a narrative review in the Introduction section. Also, we have provided the search strategy in a new paragraph in the Methods section as follows:

Three computerized electronic databases (PubMed, Web of Science, and Scopus) were searched using the following key search words: (“high-fat diet” OR “exercise programs” OR “cytokines” OR “sarcopenic” OR “FRAX” OR “BMD” OR “adipocyte” OR “inflammaging” OR “insulin resistance” OR “body composition”) AND (“osteoporosis” OR “fracture”) AND (“elderly” OR “postmenopausal”) AND (“obesity”).

2. Term 'elderly can be replaced with more inclusive words - check the recently published paper in JAGS :https://www.ncbi.nlm.nih.gov/pmc/articles/PMC6379683/

We kindly appreciate your suggestion about this issue. Accordingly, we have replaced “elderly” by “older subjects”.

3. There are a few sentences where editing should be well performed -- an example is "peoplewereincludedtoanalyzetherelationshipbetween 85"

Thank you for your advice. We have checked the manuscript and corrected this type of errors in the new version.

Round 2

Reviewer 2 Report

I found authors properly revised the manuscript according to the reviewer's points.